# CRISPR/Cas9 mutagenesis against sex pheromone biosynthesis leads to loss of female attractiveness in *Spodoptera exigua*, an insect pestt

**Shabbir Ahmed**[1], **Miltan Chandra Roy**[1], **Md. Abdullah Al Baki**[1], **Jin Kyo Jung**[2], **Daeweon Lee**[3], **Yonggyun Kim**[1] *

**1** Department of Plant Medicals, Andong National University, Andong, Korea, **2** Division of Crop Cultivation and Environment Research, Department of Central Area Crop Science, National Institute of Crop Science, Rural Development Administration, Suwon, Korea, **3** Metabolomics Research Center for Functional Materials, Kyungsung University, Busan, Korea

* hosanna@anu.ac.kr

**Data Availability Statement:** All relevant data are within the manuscript and its Supporting Information files.

## Abstract

Virgin female moths are known to release sex pheromones to attract conspecific males. Accurate sex pheromones are required for their chemical communication. Sex pheromones of *Spodoptera exigua*, a lepidopteran insect, contain unsaturated fatty acid derivatives having a double bond at the $12^{th}$ carbon position. A desaturase of *S. exigua* (*SexiDES5*) was proposed to have dual functions by forming double bonds at the $11^{th}$ and $12^{th}$ carbons to synthesize Z9,E12-tetradecedienoic acid, which could be acetylated to be a main sex pheromone component Z9,E12-tetradecenoic acetate (Z9E12-14:Ac). A deletion of *SexiDES5* using CRISPR/Cas9 was generated and inbred to obtain homozygotes. Mutant females could not produce Z9E12-14:Ac along with Z9-14:Ac and Z11-14:Ac. Subsequently, pheromone extract of mutant females did not induce a sensory signal in male antennae. They failed to induce male mating behavior including hair pencil erection and orientation. In the field, these mutant females did not attract any males while control females attracted males. These results indicate that SexiDES5 can catalyze the desaturation at the 11th and 12th positions to produce sex pheromone components in *S. exigua*. This study also suggests an application of the genome editing technology to insect pest control by generating non-attractive female moths.

## Introduction

Insect sex pheromones are volatile compounds used in many species for mate location, species recognition, and mate choice [1]. In Lepidoptera, female moths can emit species-specific sex pheromones containing unsaturated fatty acid derivatives to attract males [2]. These sex pheromones are produced by specific exocrine glands located in the 8th and 9th intersegmental membrane of moths [3, 4]. Biosynthesis of sex pheromone uses the fatty acid synthase complex

**Funding:** This work was supported by Korea Institute of Planning and Evaluation for Technology in Food, Agriculture, Forestry and Fisheries (IPET) through Exporting Promotion Technology Development Program funded by the Ministry of Agriculture, Food and Rural Affairs (MAFRA) (321100-3), Republic of Korea. The funders had no role in study design, data collection and analysis, decision to publish, or preparation of the manuscript.

**Competing interests:** The authors have declared that no competing interests exist.

to obtain palmitic acid or stearic acid as the primary substrate, which is then modified by chain-shortening via β-oxidation, added double bond(s) by catalytic activities of desaturases, and finalized by reduction or oxidation via fatty acid reductase or oxidase [5].

The beet armyworm, *Spodoptera exigua*, has sex pheromones consisting of Z9E12-tetradecadienyl acetate (Z9E12-14:Ac), Z9-tetradecen-1-ol (Z9-14:OH), Z9-tetradecenyl acetate (Z9-14:Ac), and Z9E12-tetracedien-1-ol (Z9E12-14:OH) at a ratio of 47:18:18:17 [6]. Three main components (Z9E12-14:Ac, Z9-14:OH, and Z9-14:Ac) have also been detected in Korean local populations of *S. exigua*, in which Z9E12-14:Ac can evoke more male antennal responses than others [7]. Thus, Z9E12-14:Ac is a major component of *S. exigua* sex pheromones. Double bond at the 12th carbon suggests a specific catalytic activity of Δ12 desaturase. However, due to the lack of molecular identity of Δ12 desaturase in *S. exigua* genome, it remains elusive to understand biosynthetic pathways for components of sex pheromones of *S. exigua*.

Fatty acid desaturases that introduce double bonds in specific positions of carbon chains have conserved domains such as four transmembrane domains and three histidine boxes [8]. Mutations in desaturases may lead to a reproductive isolation, which in turn causes speciation [9]. Considering species diversity in moths, desaturases associated with sex pheromone biosynthesis might have undergone a radiating evolution to generate diverse unsaturated fatty acids. Desaturases used for sex pheromone biosynthesis and confirmed by functional assays include Δ5 desaturase [10], Δ6 desaturase [11], Δ9 desaturase [12], Δ10 desaturase [13], Δ11 desaturase [14], Δ11/Δ13 dual functional desaturase [15], Δ14 desaturase [16], and a terminal desaturase [17]. Regarding Δ12 desaturase, it has been initially proposed by Jurenka [18] when assessing sex pheromone biosynthesis of two lepidopteran species (*S. exigua* and *Cadra cautella*), which commonly produce a double bond at the 12th carbon of Z9-tetradecenic acid. To find Δ12 desaturase of *S. exigua*, the transcriptome of sex pheromone gland was assessed and the most abundant desaturase named SexiDES5 was identified [19]. *SexiDES5* was clustered with Δ11 desaturase genes. Its heterologous expression in yeast can produce doubly desaturated compounds including Z9E12-14:Ac from Z9-tetradecenoic acid [20]. This suggests that SexiDES5 possesses dual functions of catalyzing both Δ11 and Δ12 desaturation and plays a main role in producing sex pheromones of *S. exigua*. To further support the physiological role of SexiDES5 in producing sex pheromones of *S. exigua*, this study used CRISPR/Cas9 (Clustered Regularly Interspaced Palindromic Repeats/CRISPR-associated protein 9) specific to gene *SexiDES5* to generate knock-out mutant females to shut down the production of sex pheromones.

## Results

### A hypothesis of dual (Δ11 and Δ12) desaturase activities of *SexiDES5*

Amino acid sequence of *SexiDES5* was clustered with Δ11 desaturases (Fig 1A). Its predicted functional domains showed typical structures of desaturase enzymes, including three histidine domains, four transmembrane domains, and one Δ9-originated catalytic domain (Fig 1B). *SexiDES5* was expressed in all developmental stages from egg stage to adult stage (except male adults) (Fig 1C). In female adults, it was specifically expressed in the abdomen containing the sex pheromone gland.

### Deletion of *SexiDES5* in the germline of *S. exigua* using CRISPR/Cas9

*SexiDES5* was located on chromosome number 23 out of 31 autosomes and one sex chromosome W of *S. exigua* genome (Fig 2A). The chromosome size was 15,261,000 bp, encoding 2,206 ORFs (https://www.biorxiv.org/content/10.1101/2019.12.26.88912v1.full.pdf). Among these putative genes, *SexiDES5* was closely linked to *mucin* and *chymotrypsin* genes. Its primary transcript contained two introns. For CRISPR mutagenesis, two sgRNAs were designed

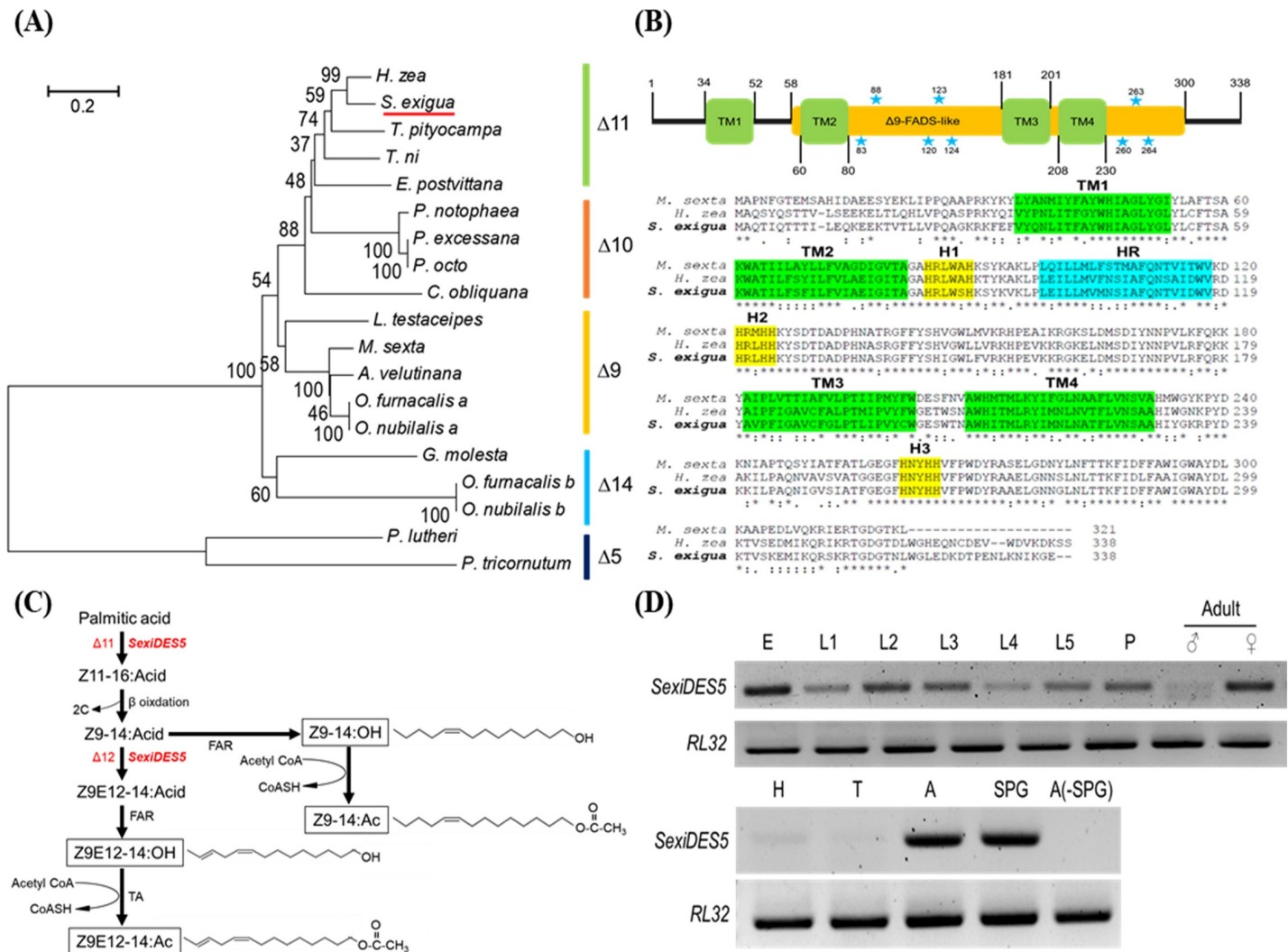

**Fig 1. Specific expression of *SexiDES5* in sex pheromone glands of adult females of *S. exigua*.** (A) A phylogenetic analysis of *SexiDES5* with other desaturases: Δ5, Δ9, Δ10, Δ11, Δ14 desaturases. GenBank accession numbers are presented in S1 Table. Bootstrapping values were obtained with 1,000 repetitions to support branching and clustering. (B) Prediction of functional domains of *SexiDES5*: histidine motifs ('H1-H3'), transmembrane helices ('TM1-TM4'), and hydrophobic region ('HR'). Δ9-originated catalytic domain and di-iron ligands are indicated as Δ9-FADS-like region with blue stars. (C) A putative biosynthetic pathway of four sex pheromone components (denoted by squares), in which SexiDES5 performs double desaturase activities at Δ11 and Δ12 positions. (D) Expression analysis of *SexiDES5* in different stages of egg ('E'), larvae ('L1- L5'), pupae ('P'), male and female adults, and in different female body parts including head ('H'), thorax ('T'), abdomen ('A'), and sex pheromone gland ('SPG'). 'A(-SPG)' represents the remaining abdomen after removing SPG. Expression of a ribosomal protein, *RL32*, confirms equal loading and the integrity of cDNA preparation.

at the third exon. After injection to embryos, the control group after injecting Cas9 without sgRNA showed an egg hatching rate of 53.7%. In contrast, the treatment group after injecting Cas9 and sgRNA showed a lower hatching rate of 34.4% (Table 1). Ultimately, the treatment group produced 61 pupae, among which 44.3% were mutants after genomic DNA sequencing. These mutants exhibited sequence deletions from 67 to 204 bp between two gRNA target sites (Fig 2B). These deletion mutants were inbred to obtain progeny ('F1', Fig 2C). PCR products of individual F1s showed wild type (single higher band), homozygote mutant (single lower band), or heterozygote mutant (double bands). Homozygote mutant were inbred to obtain the next progeny ('F2'). All these F2 individuals exhibited a single lower band indicating homozygote mutants. To confirm the segregation of mutant chromosomes, the homozygote mutant

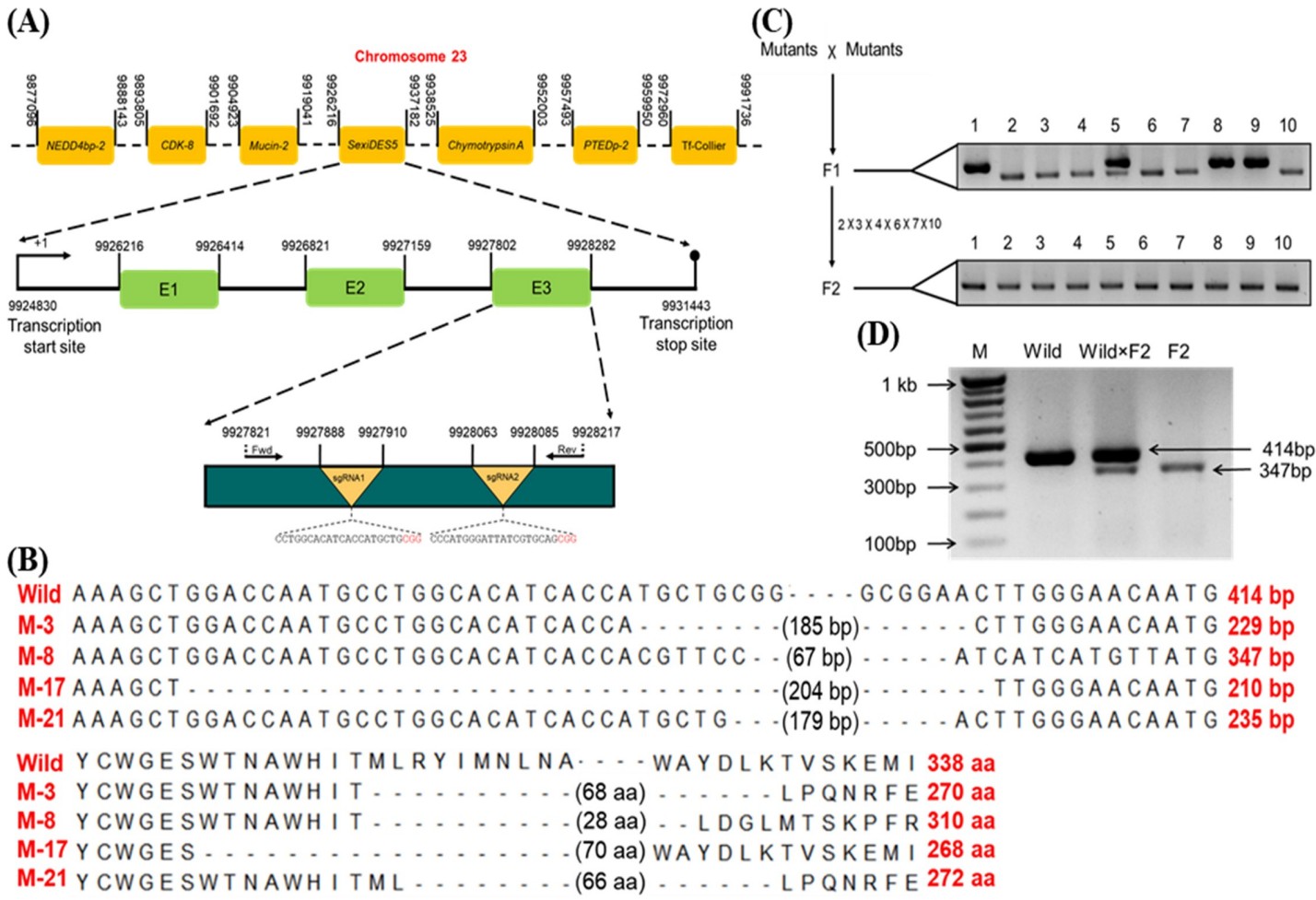

**Fig 2. Construction of a knock-out mutant of *SexiDES5* in the germline of *S. exigua* using CRISPR/Cas9.** (A) *SexiDES5* locus on chromosome 23 of *S. exigua*. 'NEDD4bp-2', 'CDK-8', 'PTEDp-2', and 'Tf-Collier' represent NEDD4 binding protein-2, cyclin dependent kinase-8, PiggyBac transposable element derived protein-2, and transcription factor collier, respectively. Three exons are presented as 'E1-E3' on a transcript from +1 to transcription stop site. Two single guide RNAs (sgRNAs) are designed in the E3, in which PAM sequences are red-colored. For sequence analysis of the CRISPR target region, two primers ('Fwd' and 'Rev') were used for PCR amplification. (B) Sequence analysis of *S. exigua* mutants (M-3, M-8, M-17, and M21) after CRISPR/Cas9 mutagenesis compared to wild type. Lengths of deletion in these mutants are denoted in parentheses in upper DNA sequences and lower amino acid sequences. (C) Selection of homozygote mutants. Inbreeding of mutants to obtain 'F1' progeny. After sequence analysis, F1 mutants (2, 3, 4, 6, 7, and 10) were inbred to obtain 'F2' progeny. (D) Confirmation of homozygote mutants (F2) by backcrossing ('Wild × F2') with wild-type individuals.

was backcrossed with wild-type individuals (Fig 2D). The hybrid progeny showed double bands for CRISPR target sites.

## Developmental characters of CRISPR mutant at *SexiDES5* (ΔSexi5)

To analyze other phenotype changes of sex pheromone biosynthesis in the mutant, growth and development of ΔSexi5 immatures were assessed. There was no significant difference in

**Table 1. Summary of *SexiDES5* mutagenesis in *S. exigua* using CRISPR/Cas9.**

| CRISPR Treatment[1] | Number of Injected Eggs | Number of Hatched Larvae | Number of Developed Pupae | Mutants |
|---|---|---|---|---|
| Control | 162 | 87 | 45 | 0 |
| Treatment | 421 | 145 | 61 | 27 |

[1] Eggs were injected with 5 nL of a mixture containing Cas9 (500 ng/μL) and sgRNA (50 ng/μL). Control eggs were injected with 5 nL of nuclease-free water.

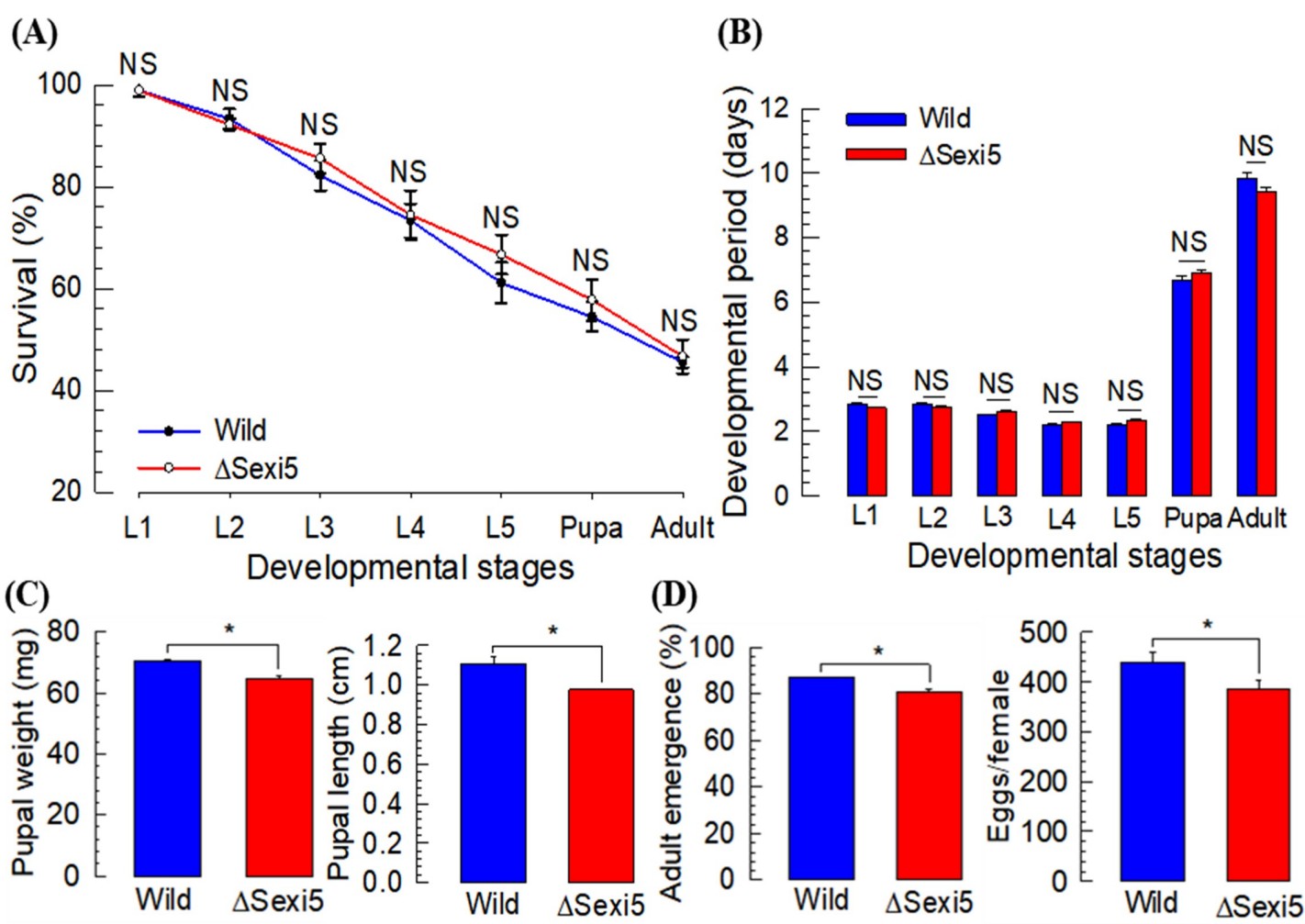

**Fig 3. Developmental profile of *SexiDES5*-deleted mutants (ΔSexi5) of *S. exigua*.** (A) Survivability in five instar (L1-L5), pupa, and adult stages of these mutants compared to the wild-type population. Each measurement (n = 10) was replicated three times. (B) Developmental rates in different stages of mutants compared to the wild-type population (n = 30). In each group, 30 samples were measured. (C) Difference in pupal body size (n = 30) on the first day after pupation. (D) Difference in adult development (n = 30) and female fecundity (n = 10) between the two groups. Asterisks (*) above standard deviation bars indicate significant difference among means at Type I error = 0.05 (LSD test). 'NS' stands for no significant difference.

survival between wild type and mutants of *S. exigua* (Fig 3A). In addition, developmental rates for all stages were not significantly different between wild type and mutant individuals (Fig 3B). However, there was a significant difference in pupal development (Fig 3C). Mutant pupae were smaller in weight and body size than wild-type pupae. The mutation also showed adverse effects on adult development, with ΔSexi5 mutant exhibiting lower emergence rate and female fecundity (Fig 3D).

## Effect of ΔSexi5 on endocrine signal for sex pheromone biosynthesis

Before analyzing sex pheromone amounts produced by ΔSexi5 mutant females, *PBAN* expression was assessed to see if such deletion mutation of ΔSexi5 might have any adverse effects on sex pheromone biosynthesis by suppressing biosynthetic endocrine signal. From the transcriptome of *S. exigua*, *PBAN* of *S. exigua* was predicted (Fig 4A). Its open reading frame had six exons (S1 Fig) encoding five different neuropeptides: diapause

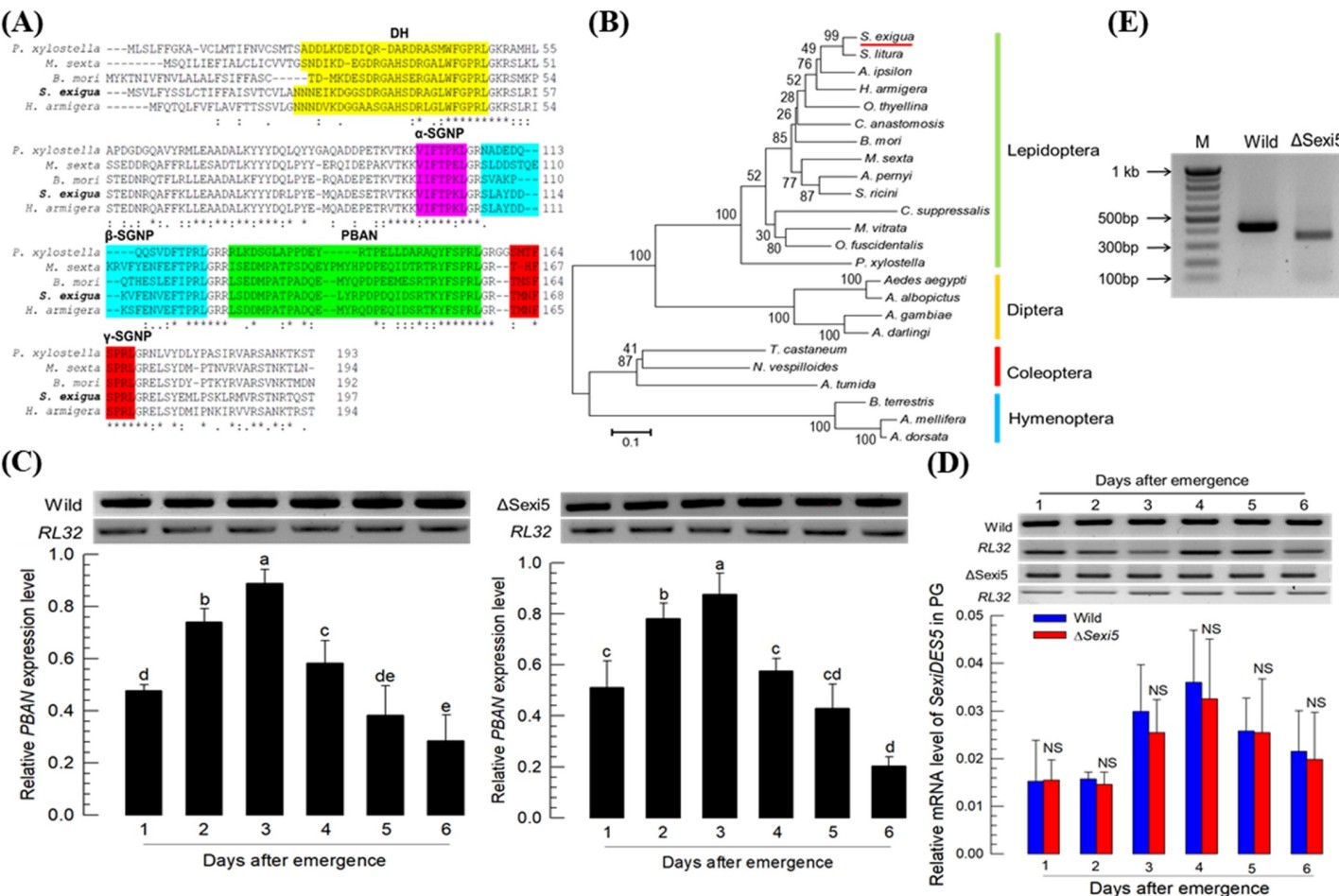

**Fig 4. Effect of a knock-out mutation of *SexiDES5* on an endocrine signal for sex pheromone biosynthesis in adult females of *S. exigua*.** (A) Domain analysis of *S. exigua* PBAN. The open reading frame of PBAN encodes five different neuropeptides: diapause hormone (DH), PBAN, α- suboesophageal ganglion neuropeptide (SGNP), β-SGNP, and γ-SGNP. (B) A phylogenetic analysis of *S. exigua PBAN* with other PBAN genes. Bootstrapping values were obtained with 1,000 repetitions to support branching and clustering. GenBank accession numbers are presented in S2 Table. (C) Expression profiles of *S. exigua PBAN* in wild-type and mutant females at different ages after adult emergence. Each measurement was replicated three times. (D) Expression profiles of *SexiDES5* in the pheromone gland (PG) of wild-type and mutant females. Different letters above standard deviation bars indicate significant difference among means at Type I error = 0.05 (LSD test) while 'NS' stands for no significant difference. (E) Difference in transcript size of *SexiDES5* between wild-type and mutant females.

hormone (DH), PBAN, α-suboesophageal ganglion neuropeptides (SGNP), β-SGNP, and γ-SGNP. Both DH and PBAN have a conserved pentapeptide ('FXPRL') at the C-terminal region. This PBAN precursor sequence of *S. exigua* was clustered with other PBAN precursors of lepidopteran insects (Fig 4B). It was found that this *PBAN* was expressed in adult females (Fig 4C). Its expression levels varied depending on adult age. Its expression level was increased after adult emergence until 3 days old and then decreased in wild type females. Such age-dependent expression regulation of *PBAN* was also observed in mutant females. *SexiDES5* was expressed in pheromone glands of mutant females (Fig 4D). Its expression also showed an age-dependent manner as seen for *PBAN* expression. There was no significant ($P > 0.05$) difference in the expression pattern of *SexiDES5* between wild type and mutant females. However, the transcript of *SexiDES5* in mutant females was shorter than that in wild-type females (Fig 4E).

## No production of sex pheromone components in ΔSexi5 females

Sex pheromone synthesis was compared between wild-type and ΔSexi5 females (Fig 5A). During the scotophase, sex pheromone glands of 2-day-old females were used to extract sex pheromone by dipping in hexane solvent to obtain volatile components. In wild-type females, three compounds (Z9-14:Ac, Z11-14:Ac, and Z9E12-14:Ac; S3 Table) of sex pheromone were detected using GC-MS. However, no compound related to sex pheromone was identified in

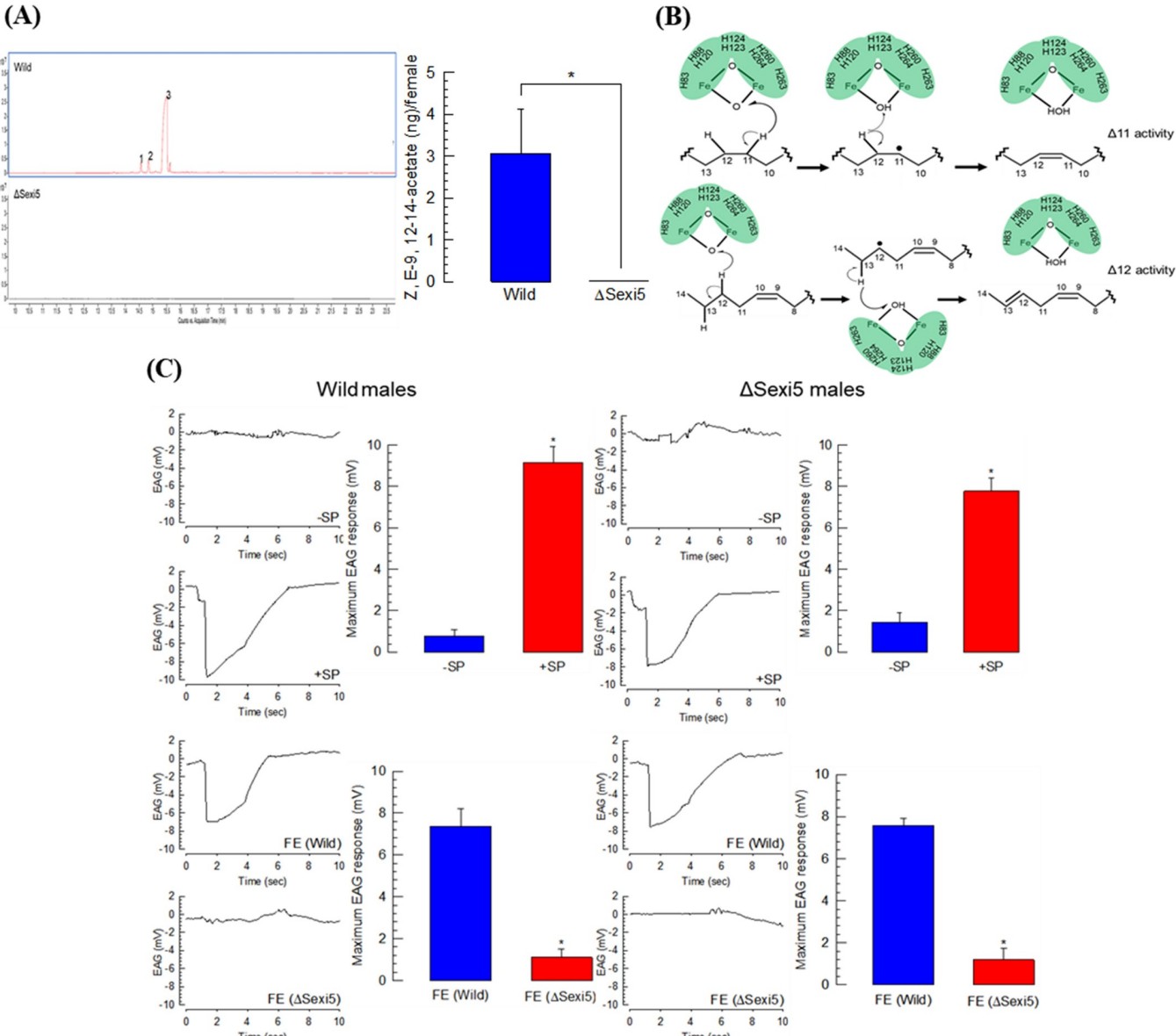

**Fig 5. Effect of a knock-out mutation of *SexiDES5* on sex pheromone biosynthesis for females and sex pheromone sensitivity for males in *S. exigua*.** (A) GC chromatograms of sex pheromone extracts of wild-type and mutant females. Three main peaks ('1–3') representing Z9-14:Ac, Z11-14:Ac, and Z9,E12-14: Ac, respectively, were confirmed by MS analysis. The major component, Z9E12-14:Ac, was quantified in both groups. Asterisks (*) above standard deviation bars indicate significant difference among means at Type I error = 0.05 (LSD test). (B) A proposed mechanism for Δ11 and Δ12 desaturase activities of Sexi5DES5. Eight histidine residues (See Fig 1B) interacting with diiron center in the active site are denoted with a green color. (C) EAG analysis to assess male sensitivities to sex pheromones in mutants of *S. exigua*. Bar diagrams were prepared using maximum EAG values for each experiment. 'SP' and 'FE' stand for sex pheromone and female extract, respectively. '-SP' stands for hexane treatment.

ΔSexi5 females at this GC detection mode. Trace amounts of compounds extracted from mutant females were assessed for the retention time zone including sex pheromone components detected in the control (S2 Fig). The retention time zone included peaks matched with different hydrocarbons such as hexadecane, heptadecane, hexadecane, hexadecane, heptacosane, pentacosane, and benzoic acid (S4 Table). Among compounds detected in the control, Z9E12-14:Ac known to be a major sex pheromone component of *S. exigua* was the main peak (3.05 ng/female) in wild-type females. However, it was not detected in ΔSexi5 females. These results support the hypothesis of dual (Δ11 and Δ12) desaturase activities of SexiDES5. This dual function was predicted using a catalytic model constructed with the crystal structure of a mammalian stearoyl CoA desaturase [21] based on the conserved active site of SexiDES5 (Fig 5B). Eight histidine residues conserved in other desaturases (see Fig 1B), could interact with oxygenated diiron center, to which an electron transport chain is formed from NADPH + H$^+$ at Δ11 for palmitic acid and at Δ12 for myristic acid to produce Z9E12-14:Acid.

ΔSexi5 males were assessed for their sensitivities to sex pheromones of *S. exigua* (Fig 5C). Both wild-type and ΔSexi5 males exhibited significant depolarization to sex pheromones in electroantennogram (EAG) analysis. Sensitivity was also observed for both wild-type and ΔSexi5 males after exposure to pheromones extracted from wild-type females. However, these males did not respond to pheromones extracted from ΔSexi5 females.

## Loss of attractiveness in ΔSexi5 females

Attractiveness of ΔSexi5 females was assessed based on mating behaviors of males (Fig 6). Males exhibited erection of hair pencil to female calling (Fig 6A). This behavior followed a diel rhythm, in which the hair pencil erection was most sensitive at the scotophase (18–24 h). At the scotophase, males were exposed to sex pheromone extracts from wild-type or ΔSexi5 females. Few males responded to sex pheromone extracts from ΔSexi5 females while most males responded well to sex pheromone extracts from wild-type females.

After checking male sensitivity to sex pheromones, their directional behavior to females was assessed using a Y-tube (Fig 6B). More than 90% males exhibited this directional behavior. Among them, more than 90% males chose wild-type females in the Y-tube. To observe their flight choice behavior, we designed an arena with a relatively large size. In this behavior assay, all males responded to females and chose wild-type females.

Finally, we assessed the attractiveness of ΔSexi5 females to males in a field near Welsh onion crop (Fig 7). Live females were placed in a Delta trap and used to monitor male catch for a week (Fig 7A). To confirm the sex, dead insects were dissected to observe developed ovaries (see photos). Field monitoring assays were performed at two seasons in five different locations (S3 Fig). While almost 160 males were caught in traps containing wild-type females, no males were attracted to traps containing ΔSexi5 females (Fig 7B). In field conditions, females lived for almost one week without showing any significant difference between wild-type and mutant groups.

Monitoring was performed at two seasons: 72020 fall (October 5 –October 30) and 2021 summer (May 15 –June 30). Live females were placed in Delta trap near Welsh onion crop and used to monitor male catch for a week. Bar graph shows the total number of captured males. Monitoring data are shown in S2 Fig. Adult lifespans of both wild-type and mutant females were measured in field conditions. Asterisks (*) above standard deviation bars indicate significant difference among means at Type I error = 0.05 (LSD test). NS' stands for no significant difference.

## Discussion

Δ12 desaturase plays a crucial role in synthesizing linoleic acid (LA) in plants and fungi [22]. Although LA is required for growth and development, most animals including insects (with

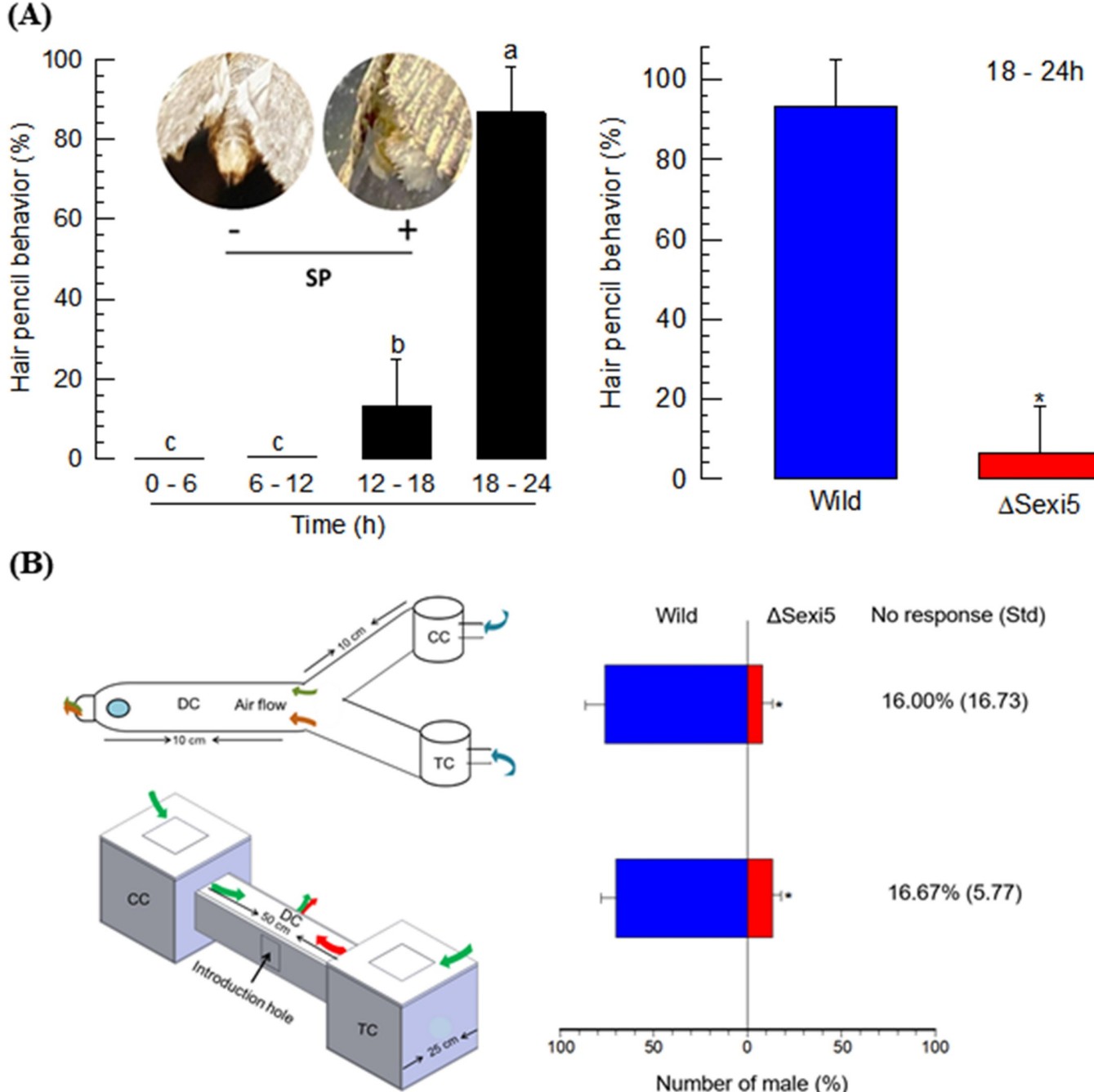

**Fig 6. Mating behaviors of males with knock-out mutant females of *S. exigua* in *SexiDES5*.** (A) Diel rhythmicity of hair pencil erection to a sex pheromone lure of *S. exigua*: light-on (0–16 h) and light-off (16–24 h). During scotophase (18–24 h), this male mating behavior was assessed after exposure to wild-type or mutant females for 30 min. (B) Male orientation behavior to females using a Y-tube or a two-way arena. 'DC', 'CC', and 'TC' stand for decision chamber containing test males, control chamber containing wild-type females, and treatment chamber containing mutant females, respectively. Each trial used 10 males. Each measurement was replicated three times. Asterisks (*) above standard deviation bars indicate significant difference among means at Type I error = 0.05 (LSD test).

some exceptions) lack Δ12 desaturase. Thus, they have to obtain this essential fatty acid through dietary means [23, 24]. Δ12 desaturases in insects such as *Acheta domesticus* and *Tribolium castaneum* have already been reported. They play a crucial role in LA production [25].

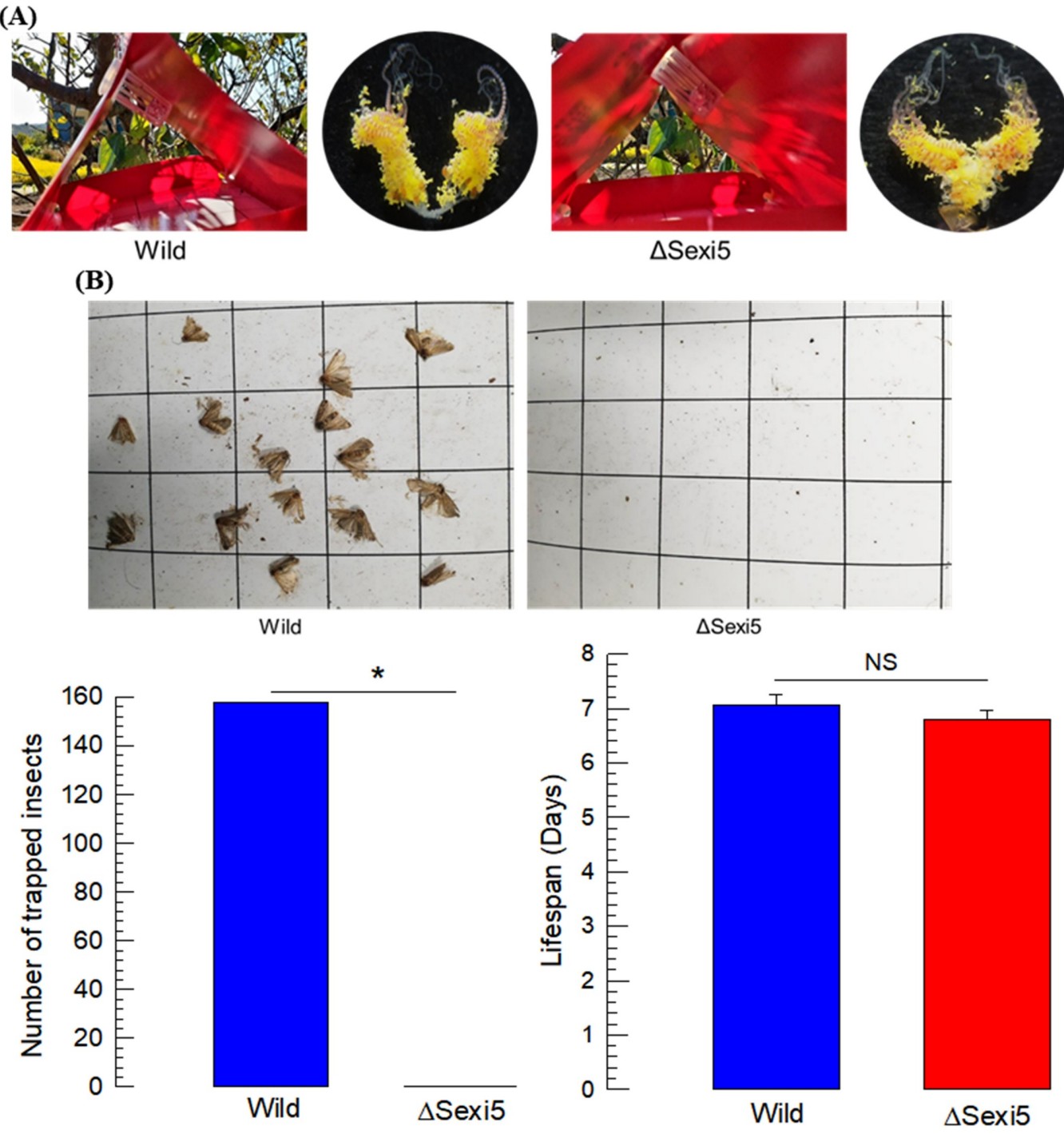

**Fig 7. Field assay of *SexiDES5*-deletion mutant females in monitoring traps.**

However, little is known in their involvement in sex pheromone biosynthesis. Sex pheromone components of lepidopteran insects include unsaturated fatty acids, which are *de novo* synthesized in their sex pheromone glands. In *S. exigua*, sex pheromone components include double bonds at the 12th position, suggesting the presence of $\Delta12$ desaturase. However, all annotated desaturase genes failed to match with a $\Delta12$ desaturase. Alternatively, Jurenka [18] has

proposed that SexiDES5 (a Δ11 desaturase) can catalyze dual desaturation at 11th and 12th carbons of a fatty acid. Our current study supports this hypothesis using its deletion mutant prepared by CRISPR/Cas9.

*SexiDES5* was specifically expressed in sex pheromone glands of adult *S. exigua*. Zhang et al. [19] have shown that transcripts of three desaturases (*SexiDES5*, *SexiDES7*, and *SexiDES11*) are highly abundant in the pheromone gland of *S. exigua*. However, *SexiDES7* and *SexiDES11* are classified as Δ9 desaturases. It is known that Δ9 desaturation is not involved in the pheromone biosynthesis of *S. exigua* [20]. *SexiDES5* is a single copy gene located on chromosome 23 of *S. exigua*. One of its nearby genes in the chromosome is a mucin, which is expressed in females to form egg chorion in ovaries [26]. This suggests that these two closely linked genes are co-expressed during the reproductive stage of females.

A deletion mutant lacking *SexiDES5* (ΔSexi5) was generated by CRISPR/Cas9, a cutting edge tool for relatively easy and efficient site-directed genome editing even in non-model insects [27]. Briefly, this system consists of two main components: Cas9 RNA-guided nuclease and sgRNA containing a 20-mer gene-specific nucleotide sequence + PAM sequence and a trans-activating CRISPR RNA sequence. Cas9 nuclease activity can lead to double-strand breaks in target DNA sequences which are then repaired by cell machinery through error-prone non-homologous end joining (NHEJ). This NHEJ approach was used in this study to prepare a mutant of *SexiDES5* of *S. exigua*. Our previous experience of preparing a mutant of $PGE_2$ receptor used one sgRNA to cut a specific target, which resulted in deletion mutants by error-prone NHEJ [28]. Our current study used two sgRNAs to remove approximately 200 bp in exon 3 of *SexiDES5*. Resulting mutants showed deletion lacking 67~204 bp between two gRNA target sites. These mutants had heterozygotes presumably containing both wild and mutant types of *SexiDES5* in homologous chromosomes. This kind of heterozygote mutant has been reported in another study for deleting an insecticide target, a ryanodine receptor gene, in *S. exigua* [29]. Thus, mutant lines were inbred to select homozygote mutants, which were confirmed by backcrossing with the wild line.

Deletion of *SexiDES5* did not influence immature developmental rate, although it showed slightly adverse effects on pupal body size and female fecundity. This might be explained by the expression of *SexiDES5* in the immature stage through association with *de novo* synthesis of LA and subsequent polyunsaturated fatty acid (PUFA). SexiDEX5 exhibiting Δ12 desaturation might catalyze the conversion of oleic acid (C18:1) to LA (C18:2). Although LA is usually supplied from diet, its additional *de novo* biosynthesis would enhance *S. exigua* development and reproduction. LA is known to be a precursor of arachidonic acid, which is the precursor of eicosanoid biosynthesis [30]. Eicosanoids and other PUFAs can mediate various physiological processes including pupal development and adult reproduction [31]. This hypothesis of *de novo* LA biosynthesis needs to be explored in *S. exigua* in the future.

There was no change in endocrine signal of ΔSexi5 mutant leading to sex pheromone biosynthesis. A PBAN receptor (PBANR) was identified from *S. exigua*. One G protein-coupled receptor shared high sequence similarity (98%) with the PBANR of *S. littoralis* [32]. In this previous study, expression levels of both *PBAN* and *PBANR* exhibited a diel rhythmicity with an increase in scotophase. This coincidence with expression levels of ligand (*PBAN*) and receptor (*PBANR*) might be interpreted as enhanced capacity of sex pheromone release and subsequent mating of *S. exigua*. Our current study showed that the increase of *PBAN* expression level varied according to adult age, with three-day-old females exhibiting the maximal level of *PBAN* expression. *SexiDES5* also showed a similar age-dependent expression pattern, suggesting the dependency of *SexiDES5* expression on PBAN. No difference in the expression pattern of SexiDES5 between wild-type and mutant females indicated that the mutation in *SexiDES5* did not influence the endocrine signal to produce sex pheromones of *S. exigua*.

ΔSexi5 mutant males responded to sex pheromones of *S. exigua*. Perception of sex phero-mones by males needs several classes of proteins including odorant-binding proteins, odorant receptors, inotropic receptors, sensory neuron membrane proteins, and odorant-degrading enzymes to generate sensory potential in olfactory receptor neurons [33]. In *Drosophila*, a spe-cific desaturase called *desat1* affects the pheromonal perception in male neurons [34]. A nor-mal sense of *S. exigua* sex pheromones in mutant males means that SexiDES5 activity is not required for the perception of sex pheromones in this species.

*SexiDES5* mutant females lost attractiveness to males. Z9E12-14:Ac, a major sex pher-omone component, was not detected in *SexiDES5* mutant while it was detected as a main peak in the sex pheromone extracted from the wild type. In addition, Z9- and Z11-14:Ac were also not produced by mutants, but produced by control females. These results reca-pitulate a role of Δ12 desaturase in sex pheromone biosynthesis. Without Δ12 desaturase ortholog, Xia et al. [20] have suggested that a Δ11 desaturase (SexiDES5) exhibits Δ11 and Δ12 dual catalytic activities. Based on this hypothesis, when palmitic acid is cata-lyzed by Δ12 desaturase activity of SexiDES5 and subsequently by β-oxidation, it would produce Z9-14:Ac. When palmitic acid is subjected to β-oxidation and subsequently cat-alyzed by Δ11 desaturase activity of SexiDES5, the main component (Z9E12-14:Ac) would be produced by Δ11 desaturase activity of SexiDES5 at the 11$^{th}$ carbon of palmitic acid, followed by chain shortening via β-oxidation and the Δ12 desaturase activity of SexiDES5 at the 12$^{th}$ carbon of myristic acid (see Fig 5B). Alternative biosynthesis path-way of Z9E12-14:Ac might be considered after chain shortening from Z11E14-14:Acid, which is produced by a combination of Δ11 and Δ14 desaturation of palmitic acid fol-lowed by chain-shortening. However, this pathway in *S. exigua* is unlikely because our current sex pheromone extract did not include Δ14 desaturase-catalytic product in our current assay. A previous transcript analysis did not find its specific or abundant expres-sion in sex pheromone gland of *S. exigua* either [19]. Furthermore in the study, heterolo-gous expression of *SexiDES5* using a yeast expression system produced Z9E12-14:Acid. This kind of unusual Δ12 desaturase activity has been experimentally demonstrated by *in vivo* labeling of pheromone biosynthesis in *Plodia interpunctella*, in which Δ11 desatur-ase catalyzes palmitic acid to form Z11-16:acid, which is β-oxidized to Z9-14:acid and then finally desaturated at the 12$^{th}$ position to form Z9E12-14:acid [35]. Similar biosyn-thesis of Z9E12-14:acid from Z9-14:acid has also been reported in the almond moth, *Cadra cautella* [18]. Thus, the deletion of *SexiDES5* by CRISPR/Cas9 failed to produce these three unsaturated fatty acids detected in the sex pheromone gland of *S. exigua* in our current assay.

Altogether, our results suggest that SexiDES5 plays a crucial role in producing sex phero-mones of *S. exigua*. Its deletion from the genome using CRISPR/Cas9 produced females lack-ing sex pheromone biosynthesis, which made them to be non-attractive to males (Fig 8). A gene drive approach [36] using CRISPR/Cas9 system against *SexiDES5* is likely to be a genetic control technique to eradicate *S. exigua* through a persistent mating disruption.

## Materials and methods

### Insect rearing

*S. exigua* larvae used in this study were originated from a Welsh onion (*Allium fistulosum* L.) field in Andong, Korea and reared for over 20 years at 25 ± 2˚C with 16L: 8D and 60 ± 5% rela-tive humidity. Five-instar (L1-L5) larvae were reared on an artificial diet [37], in which young larvae (L1 and L2) were reared in group and then separated from L3 to avoid cannibalism. Adults were supplied with 10% sucrose solution.

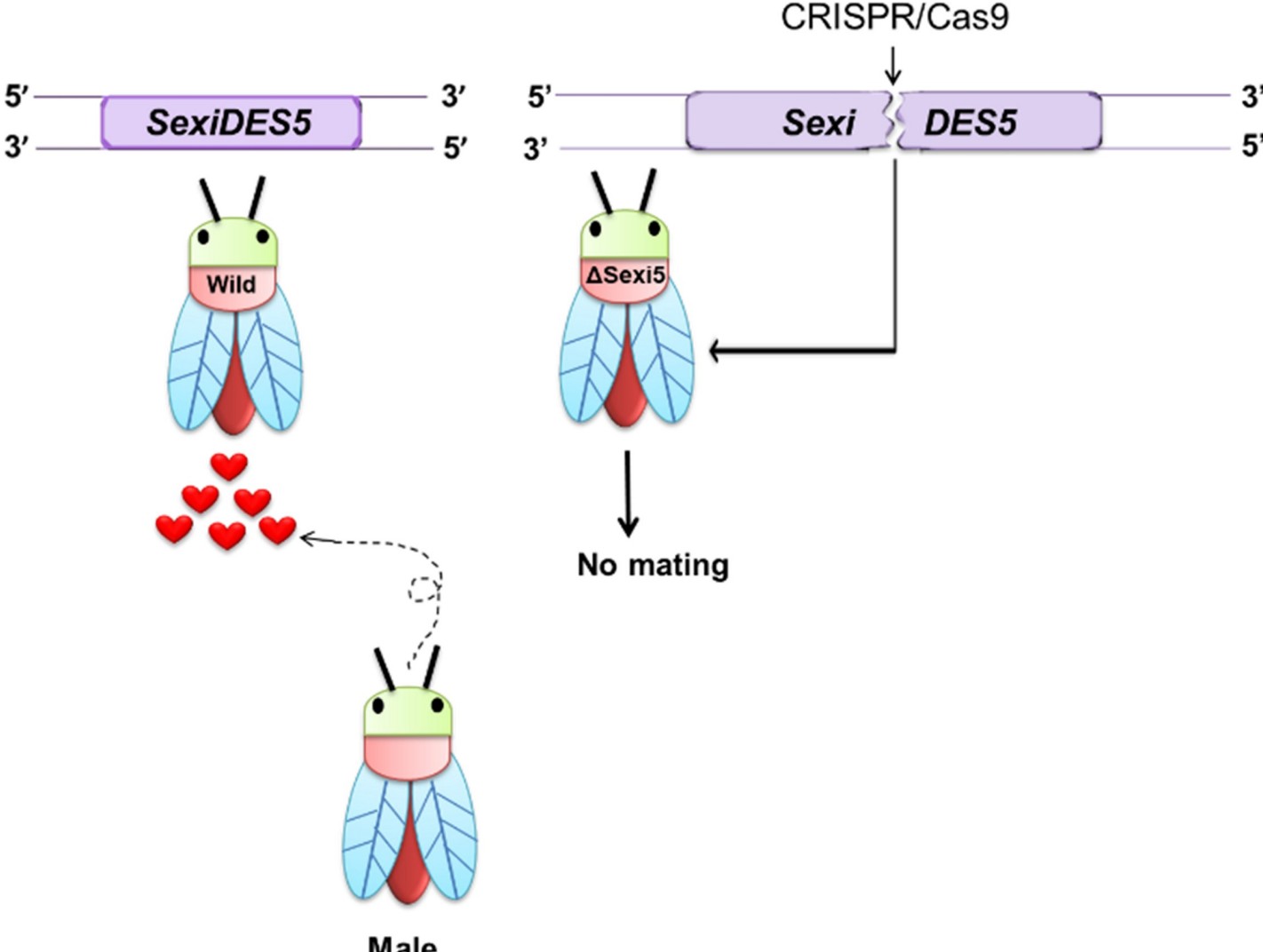

**Fig 8. A diagram demonstrating the physiological role of *SexiDES5* in producing sex pheromones in *S. exigua*.** Its knock-out mutant females generated by CRISPR/Cas9 failed to call males by losing attractiveness to males.

## Chemicals

Major (Z,E-9,12-tetradecenyl acetate) and minor (Z9-tetradecenyl acetate) components of *S. exigua* sex pheromones were purchased from ChemCruz (Santa Cruz Biotechnology, Dallas, TX, USA). n-Hexane was purchased from Sigma-Aldrich Korea (Seoul, Korea). Pheromone biosynthesis activation neuropeptide (PBAN) of *Helicoverpa assulta* and *Heliothis zea* were purchased from Bachem AG (Hauptstrasse, Bubendorf, Switzerland) and dissolved in phosphate buffer saline (PBS). PBS was prepared with 100 mM phosphoric acid plus 0.7% NaCl and adjusted to pH 7.4.

## Bioinformatics analysis

Sequences of *SexiDES5* (KU755471.1) and *PBAN* (AAT64424.1) of *S. exigua* were obtained from GenBank (https://www.ncbi.nlm.nih.gov/genbank/). Translated protein sequences were obtained with an ExPASy translation tool (http://web.expasy.org/translate/) and used for

domain analysis with SMART (http://www.smart.embl-heidelberg.de/smart/), Pfam (http://pfam.xfam.org), and Prosite (https://prosite.expasy.org/). Multiple amino acid sequence alignments were performed using ClustalW program. Phylogenetic trees were generated with the Neighbor-Joining method using the software package MEGA6.06, where evolutionary distances were computed using the Poisson correction method. Bootstrapping values were obtained with 1,000 repetitions to support branching and clustering. Chromosome 23 sequence of *S. exigua* genome was obtained from GenBank with an accession number of CM022069.1. Its gene composition was predicted using FGENESH (http://www.softberry.com/) where *Bombyx mori* genome was used as a scaffold.

## RNA extraction, RT-PCR and RT-qPCR

Total RNAs were extracted from whole bodies of all developmental stages (50 eggs, 10 L1-L2 larvae, 5 L3 larvae, 3 L4 larvae, 2 L5 larvae, 1 pupa, and 1 adult). To obtain different body parts, three adult virgin females at 2 days old were used and divided into three parts of head, thorax, and abdomen. The sex pheromone gland was obtained by cutting off five abdominal tips containing the 8th and 9th segments. RNA was extracted with Trizol reagent (Invitrogen, Carlsbad, CA, USA) according to the manufacturer's instruction. Extracted RNAs were treated with RNase-free DNase (Bioneer, Seoul, Korea). RNA quantities were determined with a Nanodrop Lite (Thermo Fisher Scientific Korea, Seoul, Korea). Then 1 μg of total RNA extract was incubated at 70˚C for 3 min and used for constructing cDNA with a RT-mix kit (Intron, Seoul, Korea). The synthesized single-stranded cDNA was used as a template for PCR amplification with gene specific primers (S5 Table) using 35 cycles of 1 min at 94˚C for denaturation, 1 min at 52˚C for annealing, and 1 min at 72˚C for extension. For checking the presence of *SexiDES5* in the developmental stages and in the different body parts of adults, primer set 1 was used (S5 Table). RT-qPCRs were performed with a qPCR machine (CFX Connect Real-Time PCR Detection System, Bio-Rad, Hercules, CA, USA) using SYBR Green Realtime PCR Master Mix (Toyobo, Osaka, Japan) with gene-specific primers of *SePBAN* or *SexiDES5* (S5 Table) according to the general guideline suggested by Bustin et al. [37]. A ribosomal protein, RL32, gene was used as a control for qPCR because a normalization using this control gene expression demonstrated relative expression of target genes in *S. exigua* [38]. Each cycle was scanned by measuring fluorescence intensity to quantify the PCR products. After the PCR reactions, melting curve analyses were performed from 60˚C to 95˚C to ensure consistency and specificity of the amplified products. Each treatment was replicated three times using independent RNA collections. Quantitative analysis of gene expression was done using the comparative CT ($2^{-\Delta\Delta CT}$) method [39]. For checking the effect of CRISPR/Cas9 mediated knocking out of SexiDES5 on truncated SexiDES5 mRNA expression, primer set 2 was used (S1 Table).

## CRISPR/Cas9—Preparation of single-stranded guide RNA (sgRNA)

Exon 3 sequence (480 bp) of *SexiDES5* was submitted to an online design tool (http://www.chopchop.com) to select two highly efficient (CRISPR efficiency scores = 62.90 and 69.82) sgRNAs (sgRNA1 and sgRNA2) containing protospacer adjacent motif (PAM) of *SexiDES5*. These two sgRNAs were separated by 175 bp. These sgRNAs were synthesized using a Guide-it sgRNA In Vitro Transcription kit (Takara Korea Biomedical, Seoul, Korea) following two steps of template DNA PCR amplification to add 20 bp target sequence and subsequent *in vitro* transcription. In the first step, PCR was performed with a customer-based forward primer (58 nucleotides = additional 4 nucleotides + T7 sequence (17 nucleotides) + target sequence (20 nucleotides) + two Gs + annealing site (15 nucleotides)) and a company-

prepared reverse primer. In the second step, sgRNAs were synthesized *in vitro* using T7 RNA polymerase according to the manufacturer's instruction. Final amounts of two sgRNAs were quantified with a NanoDrop and purified using a spin column provided with the kit.

## CRISPR/Cas9—Microinjection of sgRNA and Cas9 protein to embryos

Freshly laid eggs from 3–4 AM under darkness were dried in air for 10 min in a desiccator at room temperature. These dried eggs were fixed with a double-sided tape on a cover slip. Sharp pointed (< 20 μm diameter) glass capillaries (10 μL quartz, World Precision Instrument, Sarasota, FL, USA) were prepared with a Narishige magnetic glass microelectrode horizontal puller (model PN30, Tritech Research, Los Angeles, CA, USA) for injection. Collected eggs were injected with micro capillaries using a shutter $CO_2$ based picopump injector (PV830, World Precision Instrument) under a stereomicroscope (SZX-ILLK200, Olympus). Injection was performed with a volume of 5 nL per egg using a mixture containing Cas9 (500 ng/μL) and sgRNA (50 ng/μL) through the micropyle. All injections were accomplished within 30 min after egg collection including drying time. Treated eggs were then incubated at room temperature for 4 h before they were transferred to a growing chamber (25°C) for hatching.

## Detection of CRISPR/Cas9-mutant insects

Genomic DNAs were extracted using 5% Chelex (Biorad, Hercules, CA, USA) from exuviae of CRISPR-treated L5 larvae. Wild-type DNA was obtained from exuviae of non-treated L5 larvae. PCR conditions were: pretreatment at 98°C for 2 min; amplification step with 35 cycles of 94°C for 1 min, 52°C for 1 min, and 72°C for 1 min; and a final extension period at 72°C for 10 min. PCR products were then cloned into the pCR2.1 vector (Thermo Fisher Scientific, Seoul, Korea) and bidirectionally sequenced by Macrogen (Seoul, Korea).

## Development and survival rates at larval and pupal stages

Developmental period in each stage was defined as an elapsed time between two consecutive ecdyses. Each stage used 30 insects. Larval instars were discriminated by head capsule size as described by Goh et al. [40]. Pupal weight and length were measured on the first day after pupation. For measurement of pupal body size, 30 insects per treatment were used. Survival rate at each developmental stage was determined with three replications. Each replication used 10 insects.

## Fecundity test of adult females

Newly emerged adults were transferred to transparent insect breeding dishes (100 × 40 mm, SPL Life Sciences, Pocheon, Korea). Each dish contained one pair of male and female with a cotton wick soaked in 10% sugar solution. Newly laid eggs on surfaces of dishes were marked and recorded daily until death of the adult female. Each treatment used 10 pairs.

## Sex pheromone extraction and GC-MS analysis

During scotophase at approximately 4 h after light-off, virgin females were collected and used to extract sex pheromones. Abdominal tips (five females per replication) including the 8th and 9th segments were excised and immersed in 200 μL hexane in a 5-ml conical glass vial (Wheaton, Millville, NJ, USA) for 30 min. After removing insect tissues, the hexane extract was then transferred into another vial and used as sex pheromone extract or subsequent GC-MS analysis.

Female extracts and synthetic standards were analyzed on an Agilent 7890 GC interfaced to an Agilent 5975 mass-selective detector (Metabolomics Research Center for Functional Materials, Kyungsung University, Busan, Korea). Samples were run on a DB-WAX UI column (30 m in length, 0.25 mm in diameter, and 0.25 μm in film thicknesses, J&W Scientific, CA, USA). The oven temperature was maintained at 100˚C for 8 min, increased to 190˚C at 10˚C/min, then to 230˚C at 10˚C/min, and held for 15 min. The injection used a split ratio of 25:1. Helium was the carrier gas with a flow rate of 1 mL/min. The injector and interface temperatures were 250˚C. Electron ionization mass spectra were recorded from m/z 30 to 350 at 70 eV, with the ion source temperature of 230˚C. Quantities of each compound in female extracts were calculated using hexane solution (1 ng/μL) of synthetic sex pheromone Z9E12-14:Ac as an external standard.

## EAG analysis

A sex pheromone component (Z9E12-14:Ac; 50 ng/μL in hexane) and sex pheromone extracts from females were subjected to EAG analysis. A small sized filter paper (8 × 30 mm) soaked with 20 μL of sex pheromone component or extract was inserted into a Pasteur pipette glass as a stimulus cartridge. Male antennal response was monitored using a Syntech EAG system (Ockenfels Syntech GmbH, Kirchzarten, Germany). Antennae were obtained from adult males at 16–24 h after adult emergence. The tip of the antenna was cut and attached to the EAG electrode with an electroconductive gel (World Precision Instruments). Recording parameters of the EAG system used an air flow rate of 50 mL/min, a stimulus time of 0.1s, and an interval time of 60s. Each measurement was replicated with three different insect samples.

## Male mating behavior assay using hair pencil erection

Diel rhythmicity of hair pencil erection was measured every 6 h from light-on, at which photophase was 0–16 h and scotophase was 16–24 h. The behavior test was performed in an insect breeding dish (100 × 40 mm, SPL Life Sciences) with 10 males. After installing males and performing treatment, males showing hair pencil erection were counted for 30 min at room temperature. Each measurement was replicated three times.

## Male mating behavior assay using olfactometry

Olfactometer tests were performed using a Y-tube or a two-way box to assess the attractiveness of mutant females to males. Each Y-tube consisted of a decision chamber (10 cm) and two branches or arms (10 cm each) which terminated to the attached Teflon chambers. A constant air stream was purified with a filter of activated charcoal, heated up to 26 ± 1˚C, and humidified to a relative humidity of 70 ± 5% before it was transported through the tube system. Each chamber contained two days old wild-type or mutant females. Wild-type male moths at the same aged were released through the introduction hole and their responses to either wild or mutant female moths were observed. The time spent by males in each arm was recorded over 2 min, which began after a male crossed 75% of the Y-tube arm. Olfactometer treatments were alternated between arms every five tests to prevent locational bias. Each treatment consisted of 10 moths and replicates three times.

For a relatively large arena, two-way box test was designed to assay flight orientation. The arena consisted of one control chamber (25 × 25 × 25 cm) and a treatment chamber (25 × 25 × 25 cm). The distance between the two chambers was 50 cm. Ten wild males were released through the introduction hole to observe their orientation flight to either wild-type or mutant females. Experiments were replicated three times. Airflow was maintained at 400 mL/min by two inline flow meters (Cole Parmer Instrument, Chicago, IL, USA) for the Y-tube test and the

box assay. Observations were made for 1 h at 25 ± 2˚C with 70 ± 5% relative humidity in a dark room with 1,000 lux illumination conditions.

### Field trial of CRISPR/Cas9-mutant insects

ΔSexi5 mutant and wild-type females were used for field evaluations using a commercial Delta trap near Welsh onion fields. Pheromone traps were purchased from Green Agrotech (Kyung-san, Korea). A small plastic pheromone holder was used to hold live females. Each pheromone holder contained a single female supplied with 10% sugar solution dipped in cotton. For capturing male insects, a sticky sheet was used. Traps were hanged approximately 1.5 m above from the ground in five different localities in Andong, Korea. Traps were monitored every day up to 10 days. Sugar solution was added every day. Each trap was installed in each locality. Monitoring was performed at two seasons: 2020 fall (October 5 –October 30) and 2021 summer (May 15 –June 30).

### Statistical analysis

All studies were performed in three biologically independent replicates and plotted as mean ± standard deviation using SigmaPlot (Systat, Palo Alto, CA, USA). Means were compared by the least squared difference (LSD) test of one-way ANOVA using PROC GLM of SAS program [41] and discriminated at Type I error = 0.05.

### Supporting information

**S1 Fig. Exons of *SePBAN*.** Exons are presented as green boxes with boundary sequences. Horizontal lines between exons represent intronic regions with boundary sequences. Neuropeptides with signal peptide are presented as yellow boxes in the bottom. 'GKR', 'KK', 'GR', and 'GRR' show endoproteolytic cleavage site; 'SP', 'DH', and 'SGNP' stand for signal peptide, diapause hormone, and SGNP neuropeptides, respectively.
(TIFF)

**S2 Fig. Peaks found in GC-MS of pheromone gland extracts of wild-type and mutant females.** Peak information is given in S3 and S5 Tables.
(TIFF)

**S3 Fig. Field assay data of *SexiDES5*-deletion mutant females in attracting males.** The number of captured insects was presented as the total number from five different locations with 1.5 years (2020 full and half of 2021) of replications: Site A (October 5, 2020), Site B (October 12, 2020), Site C (October 21, 2020), Site D (May 15, 2021), and Site E (June 20, 2021). Each treatment was replicated three times. Asterisks (*) above standard deviation bars indicate significant difference among means at Type I error = 0.05 (LSD test). 'NS' stands for no significant difference.
(TIFF)

**S4 Fig. Uncropped gel pictures of Figs 1D, 2C, 2D, 4C, 4D and 4E.**
(TIFF)

**S1 Table. GenBank accession numbers used for phylogenetic analysis of desaturases.**
(TIFF)

**S2 Table. GenBank accession numbers used for phylogenetic analysis of *PBAN*s.**
(TIFF)

**S3 Table. Main peaks in the extract of sex pheromones of control females in Fig 5A.**
(TIFF)

**S4 Table. Main peaks in the extract of sex pheromones of mutant females in Fig 5A.**
(TIFF)

**S5 Table. List of PCR primers used in this study.**
(TIFF)

**S1 File.**
(PDF)

## Acknowledgments

We appreciate the design of a two-way choice test device by Fardim Sufian Roney and Heeyoon Hwang in Department of Mechanical Design and System Engineering, Andong National University, Korea.

## Author Contributions

**Conceptualization:** Yonggyun Kim.

**Data curation:** Shabbir Ahmed, Miltan Chandra Roy, Md. Abdullah Al Baki.

**Formal analysis:** Shabbir Ahmed, Miltan Chandra Roy, Md. Abdullah Al Baki.

**Funding acquisition:** Yonggyun Kim.

**Investigation:** Shabbir Ahmed, Miltan Chandra Roy, Md. Abdullah Al Baki, Yonggyun Kim.

**Methodology:** Shabbir Ahmed, Miltan Chandra Roy, Md. Abdullah Al Baki, Jin Kyo Jung, Daeweon Lee.

**Project administration:** Yonggyun Kim.

**Resources:** Yonggyun Kim.

**Software:** Miltan Chandra Roy.

**Supervision:** Yonggyun Kim.

**Validation:** Miltan Chandra Roy.

**Visualization:** Shabbir Ahmed, Miltan Chandra Roy.

**Writing – original draft:** Shabbir Ahmed, Miltan Chandra Roy, Yonggyun Kim.

**Writing – review & editing:** Jin Kyo Jung, Daeweon Lee, Yonggyun Kim.

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
