## [Decision Letter · Decision Letter 0]

1 Sep 2021

PONE-D-21-25798

CRISPR/Cas9 mutagenesis against sex pheromone biosynthesis leads to loss of female attractiveness in Spodoptera exigua, an insect pest

PLOS ONE

Dear Dr. Kim,

Thank you for submitting your manuscript to PLOS ONE. After careful consideration, we feel that it has merit but does not fully meet PLOS ONE’s publication criteria as it currently stands. Therefore, we invite you to submit a revised version of the manuscript that addresses the points raised during the review process.

We look forward to receiving your revised manuscript.

Kind regards,

Erjun Ling, Ph.D.

Academic Editor

PLOS ONE

Journal Requirements:

5. We note that Figures 6 , 7 and 8 in your submission contain copyrighted images. All PLOS content is published under the Creative Commons Attribution License (CC BY 4.0), which means that the manuscript, images, and Supporting Information files will be freely available online, and any third party is permitted to access, download, copy, distribute, and use these materials in any way, even commercially, with proper attribution. For more information, see our copyright guidelines: http://journals.plos.org/plosone/s/licenses-and-copyright.

a. You may seek permission from the original copyright holder of Figures 6 , 7 and 8 to publish the content specifically under the CC BY 4.0 license. 

Reviewers' comments:

Reviewer's Responses to Questions

**Comments to the Author**

1. Is the manuscript technically sound, and do the data support the conclusions?

Reviewer #1: Yes

Reviewer #2: Yes

2. Has the statistical analysis been performed appropriately and rigorously? 

Reviewer #1: Yes

Reviewer #2: Yes

3. Have the authors made all data underlying the findings in their manuscript fully available?

Reviewer #1: Yes

Reviewer #2: Yes

4. Is the manuscript presented in an intelligible fashion and written in standard English?

Reviewer #1: Yes

Reviewer #2: Yes

5. Review Comments to the Author

Reviewer #1: This study investigated the function of SexiDES5 in sex pheromone biosynthesis in Spodoptera exigua. This paper is well written and the data support the conclusions. It is qualified for publishing in PLOS ONE.

Reviewer #2: Sex pheromones are volatile and used to attract opposite sexes for mating in insects. In lepidopteran insects, male moths are attracted by sex pheromones from female moths which contain unsaturated fatty acid derivatives. Manuscript PONE-D-21-25798 entitled "CRISPR/Cas9 mutagenesis against sex pheromone biosynthesis leads to loss of female attractiveness in Spodoptera exigua, an insect pest" by Ahmed et al used CRISPR/Cas9 mutagenesis specific to SexiDES4 to address the physiological function and provide a new pest control tactics. The study is interesting and provides more genomic information for this species. The results will provide a basis for further analysis to reveal the underlying molecular mechanisms involved in role of SexiDES5 in sex pheromone biosynthesis. However, there are some points should be addressed by the authors before accept. I suggest this article can be published in this journal after revisions.

1. I suggest authors to better explain the reason why SexiDES5 which expressed in all the rest of tested tissues in different developmental stage.

2. Details of RT-qPCR should be added in Materials and Methods.

3. In Table S1, accession number AAT64424 should be corrected AY628764. RL32 gene was used as an internal reference, how can authors be sure that the RL32 is stable in all the samples analysed? Please add reference and the accession number in the text.

4. The manuscript is generally well written with the exception of a few spelling/grammatical errors, the English writing should be further improved.

6. PLOS authors have the option to publish the peer review history of their article (what does this mean?). If published, this will include your full peer review and any attached files.

Reviewer #1: No

Reviewer #2: No

---

## [Author Response · Author response to Decision Letter 0]

4 Oct 2021

[Reviewer #1]

This study investigated the function of SexiDES5 in sex pheromone biosynthesis in Spodoptera exigua. This paper is well written and the data support the conclusions. It is qualified for publishing in PLoS ONE.

Response: We appreciate careful review and deeply thank you for your kind encouragement.

[Reviewer #2]

Comment #2-1: I suggest authors to better explain the reason why SexiDES5 which expressed in all the rest of tested tissues in different developmental stage.

Response: We analyzed the gene expression in the whole body of larvae or pupa or adults as described this in the method section and in the figure caption (Fig. 1D). The mutant larvae and pupae showed little aberration in developmental phenotypes compared to wild type except pupal, adult developments and fecundity. The decreased body size and fecundity suggests a role of SexiDES5 other than sex pheromone biosynthesis. The dual role of SexiDES5 catalyzing desaturation suggests its catalysis to produce a PUFA, linoleic acid, which is an essential fatty acid for survival, though it is supplied from diet. We add this explanation based on speculation in the discussion as follows: “Deletion of SexiDES5 did not influence immature developmental rate, although it showed slightly adverse effects on pupal body size and female fecundity. This might be explained by the expression of SexiDES5 in the immature stage through association with de novo synthesis of LA and subsequent polyunsaturated fatty acid (PUFA). SexiDEX5 exhibiting Δ12 desaturation might catalyze the conversion of oleic acid (C18:1) to LA (C18:2). Although LA is usually supplied from diet, its additional de novo biosynthesis would enhance S. exigua development and reproduction. LA is known to be a precursor of arachidonic acid, which is the precursor of eicosanoid biosynthesis [30]. Eicosanoids and other PUFAs can mediate various physiological processes including pupal development and adult reproduction [31]. This hypothesis of de novo LA biosynthesis needs to be explored in S. exigua in the future.” 

Comment #2-2: Details of RT-qPCR should be added in Materials and Methods.

Response: We previously made a mistake of not including another primer set of SexiDES5 which was used only to check gene presence by RT-PCR. We included the primer information in S5 Table and details of RT-qPCR is added in the Materials and Methods section as follows: “For checking the presence of SexiDES5 in the developmental stages and in the different body parts of adults, primer set 1 was used (S5 Table). RT-qPCRs were performed with a qPCR machine (CFX Connect Real-Time PCR Detection System, Bio-Rad, Hercules, CA, USA) using SYBR Green Realtime PCR Master Mix (Toyobo, Osaka, Japan) with gene-specific primers of SePBAN or SexiDES5 (S5 Table) according to the general guideline suggested by Bustin et al. [37]. A ribosomal protein, RL32, gene was used as a control for qPCR because a normalization using this control gene expression demonstrated relative expression of target genes in S. exigua [38]. Each cycle was scanned by measuring fluorescence intensity to quantify the PCR products. After the PCR reactions, melting curve analyses were performed from 60oC to 95oC to ensure consistency and specificity of the amplified products. Each treatment was replicated three times using independent RNA collections. Quantitative analysis of gene expression was done using the comparative CT (2-ΔΔCT) method [39]. For checking the effect of CRISPR/Cas9 mediated knocking out of SexiDES5 on truncated SexiDES5 mRNA expression, primer set 2 was used (S1 Table).”

Comment #2-3: In Table S1, accession number AAT64424 should be corrected AY628764. RL32 gene was used as an internal reference, how can authors be sure that the RL32 is stable in all the samples analysed? Please add reference and the accession number in the text.

Response: Accession number is corrected accordingly. Reference is added in line number 339 and the accession number is added in the supplementary table S1.The use of reference gene is explained by the following explanation: “A ribosomal protein, RL32, gene was used as a control for qPCR because a normalization using this control gene expression demonstrated relative expression of target genes in S. exigua [38].”

Comment #2-4: The manuscript is generally well written with the exception of a few spelling/grammatical errors, the English writing should be further improved.

Response: Although this manuscript was cleaned by English-editing company (Harrisco, www.harrisco.co.kr), the entire text after revision is read by the corresponding author and revised at appropriate.

---

## [Decision Letter · Decision Letter 1]

18 Oct 2021

CRISPR/Cas9 mutagenesis against sex pheromone biosynthesis leads to loss of female attractiveness in Spodoptera exigua, an insect pest

PONE-D-21-25798R1

Dear Dr. Kim,

We’re pleased to inform you that your manuscript has been judged scientifically suitable for publication and will be formally accepted for publication once it meets all outstanding technical requirements.

Kind regards,

Erjun Ling, Ph.D.

Academic Editor

PLOS ONE

Additional Editor Comments (optional):

Reviewers' comments:

Reviewer's Responses to Questions

**Comments to the Author**

1. If the authors have adequately addressed your comments raised in a previous round of review and you feel that this manuscript is now acceptable for publication, you may indicate that here to bypass the “Comments to the Author” section, enter your conflict of interest statement in the “Confidential to Editor” section, and submit your "Accept" recommendation.

Reviewer #1: All comments have been addressed

Reviewer #2: (No Response)

2. Is the manuscript technically sound, and do the data support the conclusions?

Reviewer #1: Yes

Reviewer #2: (No Response)

3. Has the statistical analysis been performed appropriately and rigorously? 

Reviewer #1: Yes

Reviewer #2: (No Response)

4. Have the authors made all data underlying the findings in their manuscript fully available?

Reviewer #1: Yes

Reviewer #2: (No Response)

5. Is the manuscript presented in an intelligible fashion and written in standard English?

Reviewer #1: Yes

Reviewer #2: (No Response)

6. Review Comments to the Author

Reviewer #1: Accept. This study suggests an application of the genome editing technology to insect pest control by generating non-attractive female moths. The authors answered the questions well.

Reviewer #2: The authors addressed all the questions and comments and significantly improved the manuscripts with appropriate corrections.

7. PLOS authors have the option to publish the peer review history of their article (what does this mean?). If published, this will include your full peer review and any attached files.

Reviewer #1: No

Reviewer #2: No

---

## [Editor Report · Acceptance letter]

25 Oct 2021

PONE-D-21-25798R1 

CRISPR/Cas9 mutagenesis against sex pheromone biosynthesis leads to loss of female attractiveness in *Spodoptera exigua*, an insect pest 

Dear Dr. Kim:

I'm pleased to inform you that your manuscript has been deemed suitable for publication in PLOS ONE. Congratulations! Your manuscript is now with our production department. 

Kind regards, 

on behalf of

Dr. Erjun Ling 

Academic Editor

PLOS ONE